# A Novel Transdermal Ketoprofen Formulation Provides Effective Analgesia to Calves Undergoing Amputation Dehorning

**DOI:** 10.3390/ani10122442

**Published:** 2020-12-20

**Authors:** Paul C. Mills, Priya Ghodasara, Nana Satake, John Alawneh, Brandon Fraser, Steven Kopp, Michael McGowan

**Affiliations:** School of Veterinary Science, The University of Queensland, Gatton Campus, Gatton 4343, Australia; prg902@mail.uask.ca (P.G.); n.satake@uq.edu.au (N.S.); j.alawneh@uq.edu.au (J.A.); b.fraser@uq.edu.au (B.F.); s.kopp@uq.edu.au (S.K.); m.mcgowan@uq.edu.au (M.M.)

**Keywords:** ketoprofen, analgesia, transdermal, dehorning, calf

## Abstract

**Simple Summary:**

A transdermal formulation of ketoprofen was developed to provide effective analgesia in cattle undergoing surgical husbandry procedures. Topical administration would reduce the requirement for veterinary presence and be easy to apply to unhandled animals, improving compliance. The transdermal formulation was at least as effective as a commercially available intramuscular formulation in providing analgesia in cattle following dehorning. Efficacy was assessed using plasma cortisol concentrations, bodyweight (BW) gain and behavioural changes. Plasma ketoprofen concentrations were measured in the plasma approximately 20 min after application and peaked at 2 h, suggesting that transdermal ketoprofen had a rapid onset and would provide suitable analgesia if applied when cattle have been penned prior to the procedure being undertaken.

**Abstract:**

There is a critical need to ensure that all cattle undergoing surgical husbandry procedures are provided effective pain relief. Non-steroidal anti-inflammatory drugs (NSAIDs) are most commonly used, and typically are administered by intramuscular (IM) injection. However, administration of NSAIDs via this route to large numbers of cattle which are handled only once or twice a year, typical of many rangeland beef production systems, presents significant occupational health and safety and mis-administration risks. To address this, a novel transdermal (TD) formulation of ketoprofen was developed, and its efficacy assessed in a study of 36 Holstein–Friesian calves which were assigned to a placebo (n = 10), a TD ketoprofen (n = 10), an IM ketoprofen (n = 10) and sham dehorned group (n = 6). TD ketoprofen significantly reduced plasma cortisol concentrations between 1 to 4 h after dehorning compared to placebo treated calves, with concentrations at 2 and 4 h being very similar to those for sham dehorned calves. The expected log count of positively associated pain variables (ear flick, tail wag, ruminating, head shake, lying down, grooming and neck extending) in the TD group was reduced by 42%, compared to placebo calves, with an overall significant (*p* < 0.05) treatment effect. The IM group exhibited similar responses and both TD and IM cattle had a higher BW gain at 2 and 5 (*p* < 0.05) weeks post-dehorning, compared to placebo. This study has shown that TD administered ketoprofen was at least as effective as IM to control pain associated with dehorning and facilitates the administration of analgesic drugs prior to the surgical husbandry procedures being performed.

## 1. Introduction

Cattle are subjected to surgical procedures (dehorning, spaying and castration) as part of the routine husbandry management on beef and dairy farms [1]. It is now well recognised that these procedures cause significant acute and longer-term pain, and as such should be performed in conjunction with administration of appropriate analgesia [2]. Amputation dehorning of calves with horns attached to the frontal sinus is probably one of the most painful procedures undertaken by farmers [2,3,4].

Non-steroidal anti-inflammatory drugs (NSAIDs) can significantly reduce the pain response following surgery by limiting the inflammatory changes induced by tissue damage [2,5]. For example, ketoprofen (3 mg/kg intravenously), in combination with local anaesthetic (injected 20 min prior to the procedure), virtually eliminated the cortisol responses in 4-month-old calves castrated by one of five different methods [6]. Furthermore, systemic ketoprofen alone provided effective analgesia in calves following castration or dehorning [5,7,8,9,10]. The efficacy of ketoprofen during castration may relate to it acting on other sites, including centrally, that are not affected by local anaesthetic; plus, it has a strong anti-inflammatory effect post-operatively [2]. Similarly, the cortisol response was delayed until 5 h after dehorning, when ketoprofen but not when phenylbutazone was administered, which may relate to ketoprofen-sensitive and cortisol-sensitive sensory input [8]. Ketoprofen was also more effective than a placebo in reducing behavioural responses following dehorning in 4–8 wk old calves, which had also received a corneal nerve block [11]. Irrespective, while ketoprofen may not prevent the initial pain response during the first 1 h following dehorning, it does appear useful, and possibly superior to other NSAIDs, in ameliorating the subsequent inflammation-induced pain response [12].

NSAIDs are currently registered to be administered by intravenous, intramuscular (IM) or subcutaneous injection to ensure rapid onset of analgesia. However, injecting cattle in a handling race prior to performing surgical procedures can be associated with significant occupational health risks for farm staff and handling stress for the cattle. Thus, NSAIDs are currently often administered immediately, before or after surgical husbandry procedure(s) are performed when cattle are individually well restrained. Stafford and Mellor [6] suggested that the administration of ketoprofen intravenously at 15–20 min prior to the surgical procedure may be appropriate.

Topical NSAIDs offer an interesting alternative to other routes of administration. They can be administered with minimal expertise [13] and offer a better risk-to-benefit ratio than NSAIDs administered by other routes, including oral [14]. Topical NSAIDs, such as ketoprofen [15], have proven to be effective analgesics for acute pain in humans [16]. Unfortunately, there are no commercially available transdermal formulations containing NSAIDs registered for use in cattle. To overcome this, we have developed a topical formulation containing ketoprofen, an active already registered for use in cattle, with a pharmacokinetic profile similar to IM administration [17]. In addition to the efficacy studies cited above, ketoprofen was also selected due to a higher penetration through cattle skin in vitro, compared to other NSAIDs registered for use in cattle. The objective of the current study was to determine whether effective analgesia was induced following topical application of a transdermal formulation of ketoprofen prior to the amputation dehorning of calves.

## 2. Materials and Methods

### 2.1. Transdermal (TD) Ketoprofen Formulation

The topical formulation consisted of ketoprofen (20%) dissolved in a combination of 45%:45% ethanol and isopropyl myristate (IPM), mixed with 10% eucalyptus oil (~7% 1,8-cineole). A relatively high concentration of active was used to minimise the volume of formulation that needed to be applied, since transdermal dose rates may be several-fold higher than parenteral dose rates, depending on the active, excipients and species [13]. The placebo formulation was the same but without the active drug. A ketoprofen formulation (Ilium Ketoprofen, Troy Animal Health Care, Troy laboratories, Glendenning) registered for use in cattle was used for IM administration. A pilot study had shown that the topical formulation resulted in measurable concentrations of ketoprofen in the systemic circulation by 20 min and a maximum plasma concentration (*C*_MAX_; ~20 µg/mL) by 2 h and an area under the curve (AUC_0-Last_) of 3940 µg·min/mL. The *C*_MAX_ and AUC for IM administration were ~11 µg/mL and 2376 µg·min/mL, respectively. The bioavailability of the TD formulation was ~50% [17]. The IM results compared favourably with what had been reported previously following IM administration of ketoprofen to calves [18].

### 2.2. Animals and Treatments

This study was performed at the Queensland Animal Science Precinct (QASP) at the University of Queensland Gatton Campus and had been approved by The University of Queensland Animal Ethics Committee (SVS/455/14/MLA) by The University of Queensland Animal Ethics Committee (SVS/455/14/MLA). This approval specifically considered and approved the request to dehorn without local anaesthesia as reflecting industry practice, particularly on extensive properties.

All calves used in this study were sourced from The University of Queensland dairy. On the day before dehorning, a total of 36 weaned male Holstein–Friesian calves weighing on average 151 kg (range 86 to 236 kg) and aged 3 to 6 months were ranked according to BW, and then commencing with the four heaviest to the four lightest were randomly assigned (by blindly selecting ear tag numbers from a bag) to one of four groups:(i)Dehorning and placebo transdermal treatment (n = 10).(ii)Dehorning and TD ketoprofen (10 mg/kg; n = 10).(iii)Dehorning and IM ketoprofen (3 mg/kg; n = 10).(iv)Sham dehorning (n = 6).

As each calf presented in the crush, they were head bailed, a rail put behind their hindquarters, and then they were restrained using nose pliers to hold the head to the side. The dehorning was performed using a scoop dehorner (Bainbridge) by an experienced large animal veterinarian. Sham cattle were run through the crush but were not dehorned.

To ensure the accurate timing of blood sampling and behaviour observations, the calves were managed as a series of five replicates each containing calves from each treatment group. All ketoprofen treatments were administered 30 min before dehorning, whilst the animals were restrained in the crush. The transdermal ketoprofen or placebo (vehicle only) was applied along the backline, approximately from the shoulder to the mid-point of the backline. The treatments were applied by an operator blinded to the animal ID and wearing gloves. A 10 mL syringe was used to deliver the formulation with the nozzle used to part the hairs and apply the formulation against the skin, approximately 10 cm to the left-hand side of the spinous processes. A standard volume of 10 mL was applied to each calf. The IM ketoprofen was administered into the *gluteus medius* muscle. Each replicate of calves was subsequently brought back into the cattle handling facility and individual calves were restrained in the crush; they were head-bailed, a rail put behind their hindquarters, and then nose pliers were applied and the head was held to the side. A 10 mL blood sample was collected by jugular venepuncture (0 h) into lithium heparin vacutainer tubes, and then amputation dehorning was performed. The dehorning was performed by an experienced large animal veterinarian using a scoop dehorner (Bainbridge). This method of dehorning was used as the horns had already attached to the frontal sinus. Immediately after severing the horn, wherever possible, severed arteries were grasped with a pair of haemostats and twisted to provide effective haemostasis. Sham calves were restrained as described above and the dehorning device was laid around each horn, but they were not dehorned.

Blood samples (10 mL by jugular venepuncture following an alcohol wipe) were subsequently collected at 1, 2, 4, 8, 24, 48 and 96 h for analysis of cortisol concentration. The cattle were weighed (Truest XR3000 scales, Scintex, Brisbane, Australia, which had been calibrated immediately prior to this study) at day -1, at 2 weeks and 5 weeks after dehorning.

### 2.3. Assessment of Response to Dehorning and Analgesia

Plasma total cortisol concentration was determined for each blood sampling time using a commercially available radioimmunoassay (RIA) kit (Beckman Coulter kit IM1841, Beckman Coulter Australia, Lane Cove West, Australia) with assay sensitivity of 2.5 nM, and within assay precision of 2.7% for a quality control of 139.9 nM. Calf live weight was determined at 2 and 5 weeks after dehorning.

Calf behaviour was monitored before and after dehorning by the same person, who had been blinded to the treatments. The day before dehorning, the behaviour of all calves was monitored for 2 h to record the baseline variation, which was taken as 0 h observation. After dehorning, behaviour was monitored at 2–4 h, 4–8 h, 8–12 h, at 24 h and at 48 h. A behaviour ethogram adapted from Petherick et al. [19] was used for all observations. Twelve behaviours were monitored: head shaking, ear flicking, tail wagging, head rubbing, lying, ruminating, neck extending, grooming, walking, vocalising, feeding and drinking. Each animal was observed for 3 min at each observation period and the frequency of each behaviour was recorded. The calves were also monitored for licking the site of application and allo-grooming over the first 6 h following treatment.

### 2.4. Statistical Analysis

The behavioural changes were analysed using a confirmatory factor analysis (CFA) within the structural equation modelling (SEM) framework [20]. SEM is a conceptual map and hypothesis-driven process that is superior to principal component analysis, multiple component analysis or factorial analysis, because it permitted the use of information from all behavioural variables in the dataset to describe which ones are better predictors of another latent variable [20,21], in this case pain. The model output describes the final, and most stable models fitted using SEMSs function in STATA 13 (Stata SE for Windows, Stata Corp LLC, College Station, TX, USA). Only statistically significant behavioural variables (*p* < 0.05) remained in the final SEM model and were aggregated, and compared over time and between groups using a Poisson model. A mixed effect Poisson model with an animal fitted as random intercepts and time as random slope, and unstructured error term for the residuals. Behaviours at baseline (time 0) were centred and added to the fixed effect part of the model (essentially turning the model into a random slope model too). Experimental time was fitted as a categorical variable to derive estimates at each time point; time 2 h was used as the reference category. The analysis only considered the first 24 h, since NSAIDs are only efficacious for this direction. Treatment was also fitted as categorical variable, placebo as reference category. Overdispersion was assessed and it was ruled out as being problematic (the scale is ~1 and not significant). Overdispersed models (also those with excess zeros) were refitted using a negative binomial model. Poisson: ear flick, tail wag and negative binomial: headshake, head rub, lying, ruminate, neck extend, grooming.

Bodyweight and plasma cortisol concentrations were analysed using ANOVA.

## 3. Results

The BW of dehorned cattle administered a placebo which was significantly (*p* < 0.05) less than TD and IM cattle at 2 and 5 weeks following dehorning (Table 1 and Table 2). There were no significant differences in BW between sham-treated cattle and the TD and IM groups.

Plasma cortisol concentration increased following dehorning, as expected, with both IM and topical ketoprofen cattle having significantly lower concentrations at 1, 2, 3 and 4 h after surgery (Figure 1a,b; Table 3). However, plasma cortisol concentrations in the placebo cattle were significantly higher than the other three groups at time 0, while IM cattle were significantly higher than sham at 4 h.

For the behavioural analysis, seven behaviour variables (ear flick, tail wag, ruminating, head shake, lying down, grooming and neck extending) had substantial loading and were statistically significant (Figure 2, Appendix A). Animal with a pain score that was higher than the groups’ average had a higher Ear flick count (0.78 standard deviation [SD] higher than the average for the group), higher Tail wag (0.66 SD higher), lower Rumination (−0.36 SD) and lower in Grooming (−0.25 SD). There was also a negative correlation between Tail wagging and Lying down, and between Ear flick and Head rub, confirming that the model framework was sound and reliable. Overall, the mean counts of positively associated pain behavioural variables in the TD group were reduced by a factor of 66% (incident risk ratio [IRR] = 0.66 95% CI 0.47 to 0.92; *p* = 0.02), compared to placebo and sham group calves. The risk of higher mean counts of positively associated pain variables was observed at 2 h and 4 h after dehorning.

## 4. Discussion

This study has shown that a topically applied formulation containing ketoprofen could deliver effective analgesia in cattle undergoing dehorning. A range of parameters were used to compare efficacy, including cortisol, BW and behaviour. In addition, similar efficacy was shown between transdermal and IM ketoprofen, which was not surprising, since a pilot study had shown somewhat higher plasma concentration time curves for TD [17]. However, this can easily be managed by lowering the dose rate, while the prolonged plasma drug concentrations would contribute to efficacy following a single application. What was surprising from the pilot study [17] was how rapidly ketoprofen reached the systemic circulation following topical administration, with a lag time (~20 min) rivalling IM administration.

This study was performed on calves at an age when they would normally be dehorned. A cornual nerve block was not used (this study plan was approved by the UQ animal ethics committee), since this represents the industry standard currently. A sham-handled group was used as a negative control to allow for the stress induced by handling and multiple blood samples collected. The remaining calves were all dehorned and no adverse effects were observed, either from the procedure or the drug administration. There was potentially a higher risk of adverse effects commonly associated with NSAIDs (i.e., renal, gastrointestinal and haemostatic) following TD administration due the higher maximum plasma concentration, compared to IM, although this was considered minimal, since only a single dose was used.

There are concerns over the validity of using plasma cortisol concentrations to monitor pain and distress [22,23]. One of these concerns is the high variability in resting cortisol levels and the response to stress. For example, mild stress (confining in a pen) induced significant differences in the cortisol responses within a group of nine Angus/Hereford cows [24]. It is therefore prudent to use other indicators of stress and, particularly for the current study, pain when monitoring bovine responses. However, the changes in plasma cortisol concentrations found in the current study reflect what has been reported in earlier studies [11,25,26]. Stafford and Mellor [2] describe a sharp increase in cortisol during the first 1 h following surgical dehorning, which is due to the pain response and not responsive to NSAIDs. Similarly, ketoprofen (IV 15–20 min prior) had little effect on 3–4-month-old calves on the initial cortisol peak, but the mean plasma cortisol concentrations were similar to non-dehorned controls from 2–9.5 h [11]. The subsequent inflammatory response was significantly responsive to ketoprofen, particularly by transdermal administration, during the first 4 h, with a visible but not significant reduction over the following 24 h. Plasma cortisol concentrations generally return to pre-treatment levels by 9 h, which the placebo group in the current study appeared to do. However, a small but insignificant rise is the other three groups could not be explained. Despite these findings, greater cattle numbers would have been useful to increase the power of the study, particularly when considering the significant inter-individual variability.

Behavioural responses are used to monitor pain and distress in animals [4,27]. An early study reported that ketoprofen appeared to offer only a small reduction in adverse behaviours [11]. In the current study, a behavioural ethogram adapted from a previous study [19] was used to provide an objective outcome for any changes in behaviour. Again, more animals in each group would have improved the statistical significances (statistical power = 17%) calculated. There were seven variables (ear flick, tail wag, ruminating, head shake, lying down, grooming and neck extending) that had a significant and negative association with pain. Importantly, there was a significant overall positive response to ketoprofen analgesia, supporting the notion that inflammation is a major contributor to pain after the first hour following surgical dehorning [2].

Appetite and feed intake also form part of the behavioural response to pain [28,29]. However, subsequent changes in BW, or lack thereof, are objective and definitive outcomes to surgical pain and, indeed, loss of productivity is a major indication for the use of analgesia when dehorning cattle [4,27]. In the current study, surgical dehorning resulted in a significantly lower BW gain by 2 and 5 weeks (*p* < 0.05) after the procedure. Importantly, BW gains alone would more than justify the cost of administering analgesia, particularly if this was by a non-veterinarian and avoiding call-out fees.

It could be expected that the use of a NSAID would ameliorate the physiological and behavioural responses to surgical pain. The initial pain response may be unchanged, but certainly cortisol responses are not as pronounced by 2 h after dehorning if ketoprofen is used [11]. Interestingly, a similar response was not found when phenylbutazone was administered, suggesting a potential central effect of ketoprofen, in addition to the primary anti-inflammatory effect [8]. The major outcome from the current study was that a comparable analgesia to IM ketoprofen could be achieved using a topically administered formulation. Indeed, the transdermal formulation appeared to be slightly superior to an IM administration, although this probably relates to a similar lag time, but a higher and prolonged area-under-the-curve (AUC) for the transdermal formulation, indicating a slightly higher overall drug exposure. Allo-grooming was not observed during the 6 h following treatment, so it would appear to systemic drug concentrations were primarily due to transdermal penetration of ketoprofen. The lack of statistical differences is related to the lack of statistical power in this study. To clarify, assuming a power of 80%, significance level percentage of 95%, balanced groups size, a probability of observing an outcome in the control group of 80%, the required sample size for this study to detect an odd ratio (or IRR) of 4 or greater is 88 animals in each group, with the current study having a power of 17%.

## 5. Conclusions

Administration of a transdermal formulation containing ketoprofen to calves prior to amputation dehorning resulted in a rapid onset of analgesia, very similar in duration and efficacy to that induced by IM administration of ketoprofen. Because of the recognised ease of administration of transdermal formulations to cattle, these findings are likely to facilitate greater routine use of analgesia in cattle undergoing painful husbandry procedures such as dehorning.

## Figures and Tables

**Figure 1 animals-10-02442-f001:**
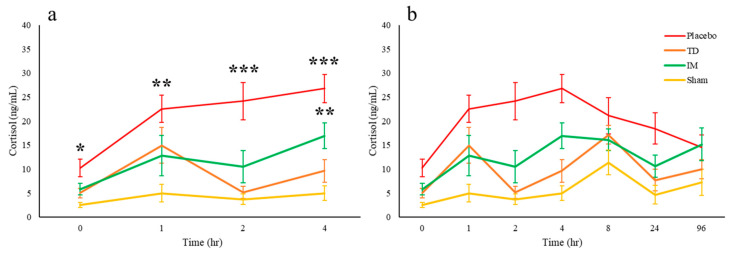
Plasma cortisol concentrations (ng/mL) changes (mean ± SD) at 0–4 h (**a**) then to 96 h (**b**) after dehorning in cattle treated with placebo (n = 10), TD ketoprofen (n = 10; 10 mg/kg), IM ketoprofen (n = 10; 3 mg/kg), compared to a sham group that were not dehorned (n = 6). Significant differences of *p* < 0.05 (*), *p* < 0.01 (**) or *p* < 0.001 (***).

**Figure 2 animals-10-02442-f002:**
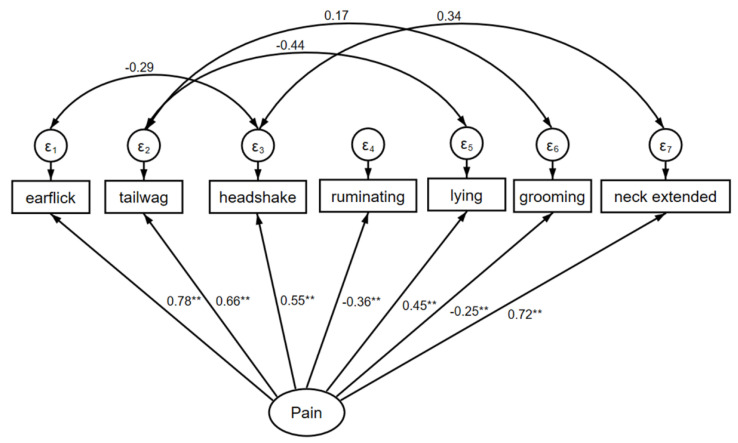
Line plot showing Structural Equation Modelling and Confirmatory Factor Analysis conceptual map depicting the association between calves’ behavioural variables (fitted as measured predictor variables) and pain response (fitted as a latent variable). ** *p* < 0.001.

**Table 1 animals-10-02442-t001:** Summary table of bodyweight (BW; kg) changes (mean ± SD) at 2 and 5 weeks after dehorning in cattle treated with placebo (n = 10), transdermal (TD) ketoprofen (n = 10; 10 mg/kg), intramuscular (IM) ketoprofen (n = 10; 3 mg/kg), compared to a sham group that were not dehorned (n = 6).

Group	0 (Dehorned)	2 Weeks	5 Weeks
Placebo	151.7 ± 41.6	175.9 ± 38.8 *	205.8 ± 43.8 *
TD ketoprofen	151.9 ± 43.5	181.2 ± 49.2	212.8 ± 54.4
IM ketoprofen	151.8 ± 46.4	178.4 ± 50.9	212.5 ± 55.2
Sham	154.2 ± 49.9	181.2 ± 53.4	215.1 ± 59.2

* Significantly lower (*p* < 0.05) than TD, IM and Sham.

**Table 2 animals-10-02442-t002:** Effects of treatment group on calves’ bodyweight (BW) changes at Days 0, 14 and 35 after dehorning in cattle treated with placebo, n = 10; transdermal (TD) ketoprofen, n = 10; intramuscular (IM) ketoprofen, n = 10; compared to a sham group, n = 6 that were not dehorned. Values in rows are means, which were analysed using ANOVA, with the outcomes for treatment, time and treatment * time indicated.

Treatment Group	Mean × (kg)	Day				*p* Values		
		0	14	35	SEM (kg)	Treatment	Time	Treatment × Time
Placebo	177.8	151.7	175.9	205.8	8.4	**0.83**	**<0.001**	**0.16**
TD ketoprofen	182.0	151.9	181.2	212.8	9.8			
IM ketoprofen	180.9	151.8	178.4	212.5	10.1			
Sham control	183.4	154.2	181.2	215.0	13.5			

**Table 3 animals-10-02442-t003:** Effects of treatment group (placebo, n = 10; TD ketoprofen, n = 10; IM ketoprofen, n = 10; sham, n = 6) on calves’ plasma cortisol concentration changes following dehorning or control. Values in rows are means, which were analysed using ANOVA, with the outcomes for treatment, time and treatment * time indicated.

Treatment Group	Mean (ng/mL)	Time (h)								*p* Values		
		0	1	2	4	8	24	96	SEM (ng/mL)	Treatment	Time	Treatment * Time
Placebo	19.7	10.2	22.5	24.2	26.8	21.1	18.5	14.5	1.3	**<0.001**	**0.02**	**0.16**
TD ketoprofen	9.8	5.0	15.0	5.2	9.7	17.1	7.7	9.9	0.9			
IM ketoprofen	6.2	2.6	5.0	3.7	5.0	11.4	4.7	7.3	0.6			
Sham control	12.6	5.8	12.8	10.5	17.0	16.1	10.6	15.2	1.1			

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
