# Peer review of "A Novel Transdermal Ketoprofen Formulation Provides Effective Analgesia to Calves Undergoing Amputation Dehorning"

_animals, 2020, doi:10.3390/ani10122442_

Round 1

Reviewer 1 Report

A Novel Transdermal Ketoprofen Formulation 2 Provides Effective Analgesia to Cattle Undergoing 3 Amputation Dehorning

Review – animals – 1022559

Feedback to the authors.

General comments

Dear Authors, I found your paper very interesting and your research question of importance. I am very happy to see that people are tackling the welfare of animals in the meat/dairy industry and are looking at ways to make it better for the animals and more manageable for the farmers.

Throughout the papers, I found some inconsistencies on how you expressed the word “hours”. Sometimes it is the word and sometimes “h” or “hrs”. Please be consistent.

I have the same comment for the word “bodyweight” vs “BW”.

Simple summary

No comments

Abstract

Line 24-25: “only once or twice a year” instead of “one or two times” might be better

Line 36: the authors mention that the study showed that TD ketoprofen facilitates the administration of pain relief. After reading the paper, I did not see any data confirming that information. It is pure speculation. I agree that we would expect TD to be more comfortable than IM, but the current study did not demonstrate that.

Introduction

The introduction lacks specificity in regards to the actual paper. The authors should focus on dehorning, pain associated with it and the efficacy of NSAIDs in dehorning associated surgery. It would be more logical and would bring more weight and explain better why the authors have chosen to look at ketoprofen TD in calves during dehorning procedures. Besides, it would prepare the readers better for the discussion of the results.

Line 44-46: Although both your references are correct, I would be tempted to add on a more recent one: Stafford and Mellor (2011, “Addressing the pain associated with disbudding and dehorning in cattle”). This reference is more specific to your question than the Stafford Mellor 2009, which is too general.

Line 48 – 52: While I understand why you are speaking about the effectiveness of the use of NSAID's in providing analgesia during/post-surgery, I wonder why you used references about castration while we have plenty of references on dehorning and analgesia in the literature. Also, the pain for castration is very unlikely similar to the one associated with dehorning. And different types of dehorning cause different types of trauma and pain. I wonder if it would bring more strength to your paper if you would speak about the pain associated with dehorning (different methods), the effect of NSAID’s on dehorning. It would explain better why you decided to develop in a new methodology for a specific NSAID’s and why you looked at it the way you did (i.e. type of assessment).

Line 52: reference #7 is not about castration but dehorning only.

Line 52-53: the authors introduce ketoprofen use for castration. Again I think this is a poor choice. We have enough references in the literature about dehorning and the use of NSAID's and Ketoprofen in particular (Stafford et Mellor, 2011; Sutherland et al, 2002; McMeekan et al., 1998; Espinoza et al., 2016; Stock et al., 2016 (carprofen); Faulkner & Weary, 2000; Milligan et al., 2004. I think those should be used to introduce your work and bring more strength to your work.

Line 54 – 59: this part feels very rushed. The authors introduce the notion of delayed analgesia that we see during dehorning when using an NSAID and ketoprofen in particular. This aspect is essential for your paper, and good development would help later on in the Discussion section. It needs to be redeveloped and better explained.

Line 70: Although there might not be any commercially available  TD formulation, the authors should at least acknowledge other groups that looked at the use of TD formulation of NSAID’s for dehorning: Kleinheinz, 2016, “Effects of transdermal flunixin meglumine on pain biomarkers at dehorning in calves”.

Line 74-75: I think we need a reference regarding the fact that ketoprofen has a high penetration in cattle compared to others.

Materials and Methods

Line 86 – 91: “A pilot study….was 50%”. This section should be in the Introduction section if anywhere.

Line 91-92: “The IM results…to calves (16)”. I am not sure what this sentence means.

Line 93-94: Was the TD ketoprofen or placebo put by hand or by using an applicator? Were people wearing gloves? Was the TD solution set against hairs direction to ensure dermal contact, or it did not matter? Is the backline the spine? What quantity? I think the readers need more explanation regarding the actual procedure. Remember that the Mat and Method section should be descriptive enough to allow the reader to reproduce the experiment.

Line 124 – 126: Blood samples. How many millilitres of blood were taken at a time? From which vein? Did you put a jugular catheter? Was it the tail vein? Preparation?

Line 132: was the assessor blinded? Needs to be specified?

Line 135: a reference for the ethogram?

Line 152: “was” should be “were”.

Table 2: but for the title, there is no explanation regarding the table itself. What is the table about? Remember that Tables and Figures should be able to stand alone.

Figure 1: Do you need two figures here? I believe 1b is clear enough to present your data.

Line 176: 96 hours not weeks

Line 179: Table 2 should be table 3

Table 3: same comment as for Table 2

Line 180: “variables”. Shouldn't it be “behaviours” for consistency?

Line 182 – 185: I understand what you mean, but the sentence is difficult to read. Is there any way it could be simplified?

Lines 187, 189 and 191: Placebo or placebo? Please be consistent throughout the paper

As a general comment, I have the impression that the interpretation of the statistical results has been written by a statistician, not a clinician. I found it very difficult to read and to understand. It would be useful to develop the interpretation (using examples) to increase comprehension.

Figure2: same comments as for tables 2 and 3.

Discussion

Line 198: but for cortisol, BW and behaviours, what else did you measure? That sentence implies you measured something else? I could not find what else? Please rewrite.

Line 201: a reference for the pilot study is missing

Line 203-204: The authors refer again to results from the pilot study. It is not clear at all. We need more explanation and reference.

Line 212-225: Paragraph on cortisol. It is a very poorly written paragraph. The readers need a better description of the curve for cortisol in other studies, followed by a comparison with the current study and a conclusion.

Line 227 to 235: Very poorly written paragraph. I do not understand what the authors are demonstrating. First of all, to prove that study from McMeekan et al. (1998) is comparable, the authors should explain that it was done in calves of the same age with a similar technique (scoop). Second, if the authors had paid attention to that study, they would have realised that McMeekan at eal did not look at any behaviours, but only at the plasma concentration of cortisol. Therefore, Line 227-228 is misleading.

Line231: why is the power not part of the Mat & methods and Results sections?

Line 229 to 230: if I am correct, the ethogram consists of several observations and is therefore depending on the observers’ interpretations. Subjectivity is, consequently, a possible issue. The authors say that their adapted ethogram provides objective values. I don’t think this is correct as no validity tests were performed to prove this (i.e. interrater; interrater). It would have been better if more than one person had assessed the behaviours.

Line 233-235: This sentence does not belong here.

Line 237 – 243: very poorly written paragraph.

Line 240-241: this statement is only true for Sham, TD and IM, not for the placebo group.

Line 245-247: Very poorly written. It needs rewriting. Moreover, the authors are again misleading the readers with, in this case, the study from Sutherland and Mellord (2002, ref #7). I agree that the authors of that study looked at central vs peripheric effects. But they concluded that most of the effect of ketoprofen were through the anti-inflammatory effect in the periphery and that the central effect, if any, was of very little importance. It is the opposite of what the authors in the current study seem to imply.

Line 255-259: This should be part of the results section.

Line 245-259: very poorly written. It needs to be developed better and rewritten.

Conclusion:

While info on line 261 could be extrapolated from the results of this study, line 262-263 was not part of the study. Nowhere in the study, we looked at the facility of providing the TD to cattle, even fractious ones. It is pure speculation.

In summary, the discussion section lacks of information on the different types of dehorning methods and how they compare with the one used in the study. There is no discussions about the different ways of assessing pain in calves, objective vs subjective and how it compares with this study.

There is no real discussion of the weakness of the papers: the validity of the assessment method, the use of only one observer etc.

In summary:

I think you have a good study that deserves to be published. Still, the authors must rework on the introduction, the presentation of the results and most notably on the structure and development of the discussion.

My main concern about your results (and the design of the study) is that you only used one assessor. You should have filmed the calves and have different people assessing the behaviours. It would have given you better quality data. Moreover, you could have used the results to validate your assessment method partially.

Thank you

Author Response

A novel transdermal ketoprofen formulation provides effective analgesia to cattle undergoing amputation dehorning – response to reviewer’s comments

Reviewer #1

General comments

Dear Authors, I found your paper very interesting and your research question of importance. I am very happy to see that people are tackling the welfare of animals in the meat/dairy industry and are looking at ways to make it better for the animals and more manageable for the farmers.

Throughout the papers, I found some inconsistencies on how you expressed the word “hours”. Sometimes it is the word and sometimes “h” or “hrs”. Please be consistent.

I have the same comment for the word “bodyweight” vs “BW”.

These have all been changed to hr or BW, respectively, except the first use of bodyweight (BW) to define the abbreviation.

Simple summary

No comments

Abstract

Line 24-25: “only once or twice a year” instead of “one or two times” might be better

Changed

Line 36: the authors mention that the study showed that TD ketoprofen facilitates the administration of pain relief. After reading the paper, I did not see any data confirming that information. It is pure speculation. I agree that we would expect TD to be more comfortable than IM, but the current study did not demonstrate that.

‘pain relief’ was changed to ‘analgesic drugs’ but the emphasis was that it was easier to apply and topical administration is certainly easier than IM administration.

Introduction

The introduction lacks specificity in regards to the actual paper. The authors should focus on dehorning, pain associated with it and the efficacy of NSAIDs in dehorning associated surgery. It would be more logical and would bring more weight and explain better why the authors have chosen to look at ketoprofen TD in calves during dehorning procedures. Besides, it would prepare the readers better for the discussion of the results.

We agree that a greater focus on dehorning would be useful, so have added two references (Duffield et al 2010; Stafford and Mellor, 2011).  However, the focus of the paper is primarily that a TD formulation of ketoprofen successfully penetrated bovine skin and achieved systemic concentration of active drug greater than IM administration.  We were not intending to re-do the great work that others, particularly Stafford and Mellor, had already done to show that dehorning was painful and required suitable analgesia. We were particularly interested to demonstrate that analgesia can be applied in an extensive property to unhandled stock.

Line 44-46: Although both your references are correct, I would be tempted to add on a more recent one: Stafford and Mellor (2011, “Addressing the pain associated with disbudding and dehorning in cattle”). This reference is more specific to your question than the Stafford Mellor 2009, which is too general.

Done

Line 48 – 52: While I understand why you are speaking about the effectiveness of the use of NSAID's in providing analgesia during/post-surgery, I wonder why you used references about castration while we have plenty of references on dehorning and analgesia in the literature. Also, the pain for castration is very unlikely similar to the one associated with dehorning. And different types of dehorning cause different types of trauma and pain. I wonder if it would bring more strength to your paper if you would speak about the pain associated with dehorning (different methods), the effect of NSAID’s on dehorning. It would explain better why you decided to develop in a new methodology for a specific NSAID’s and why you looked at it the way you did (i.e. type of assessment).

As noted above, we have added more information about the pain associated with dehorning. As we also noted above, we were not trying to re-do the excellent studies already done, but to show that a topical administration can provide systemic drug concentrations at least as high as IM administration, but substantially easier to apply (i.e. by non-veterinary staff).  Moreover, the fact that systemic ketoprofen concentrations attained or exceeded IM administration is basically strong support for efficacy (bioequivalence), but we wished to also demonstrate efficacy by traditional indices. Furthermore, the castration studies appeared to raise the issue of a possible central action of ketoprofen more effectively.

Line 52: reference #7 is not about castration but dehorning only.

‘or dehorning’ was added to the sentence

Line 52-53: the authors introduce ketoprofen use for castration. Again I think this is a poor choice. We have enough references in the literature about dehorning and the use of NSAID's and Ketoprofen in particular (Stafford et Mellor, 2011; Sutherland et al, 2002; McMeekan et al., 1998; Espinoza et al., 2016; Stock et al., 2016 (carprofen); Faulkner & Weary, 2000; Milligan et al., 2004. I think those should be used to introduce your work and bring more strength to your work.

We have already added Stafford and Mellor, 2011; and also Duffield et al 2010, while Sutherland et al 2002 was already there and McMeekan et al 1998 was introduced a little later specifically for its link with cortisol.  As noted earlier, the efficacy of ketoprofen following castration was also to highlight a potential central effect of ketoprofen. Importantly, we consider that this study was specifically about the success in administering ketoprofen transdermally, using dehorning as a model, rather than determining the specific efficacy of ketoprofen to provide analgesia following dehorning.

Line 54 – 59: this part feels very rushed. The authors introduce the notion of delayed analgesia that we see during dehorning when using an NSAID and ketoprofen in particular. This aspect is essential for your paper, and good development would help later on in the Discussion section. It needs to be redeveloped and better explained.

This was intended as a continuation of the fact that ketoprofen may be superior to other NSAIDs, and hence part of the reason it was selected, so the following was added into this sentence: ‘and possibly superior to other NSAIDs’

Line 70: Although there might not be any commercially available  TD formulation, the authors should at least acknowledge other groups that looked at the use of TD formulation of NSAID’s for dehorning: Kleinheinz, 2016, “Effects of transdermal flunixin meglumine on pain biomarkers at dehorning in calves”.

We agree that this formulation is available elsewhere and, indeed, registered in some countries.  However, the specifics (particularly the excipients) of the formulation have not been made public and the authors of these studies have never produced data regarding systemic concentrations of the active.  Of concern, this formulation was only initially (and still continues to our knowledge) to have a claim against pyrexia and not inflammation.  Until more information is available, we have avoided reference to this formulation.  As a side note, our in vitro studies showed that we could obtain transdermal penetration of flunixin, although ketoprofen was superior.

Line 74-75: I think we need a reference regarding the fact that ketoprofen has a high penetration in cattle compared to others.

Done

Materials and Methods

Line 86 – 91: “A pilot study….was 50%”. This section should be in the Introduction section if anywhere.

Done

Line 91-92: “The IM results…to calves (16)”. I am not sure what this sentence means.

This line was removed and ‘…, which was similar to what had been reported previously following IM administration [16]’ was added to the sentence on pharmacokinetics on line 90.

Line 93-94: Was the TD ketoprofen or placebo put by hand or by using an applicator? Were people wearing gloves? Was the TD solution set against hairs direction to ensure dermal contact, or it did not matter? Is the backline the spine? What quantity? I think the readers need more explanation regarding the actual procedure. Remember that the Mat and Method section should be descriptive enough to allow the reader to reproduce the experiment.

The following was added: ‘This was performed by an operator blinded to the treatment and wearing gloves. A 10 mL syringe was used to deliver the formulation with the nozzle used to part the hairs and apply the formulation against the skin, approximately 10 cm to the left hand side of the spinous processes.’

Line 124 – 126: Blood samples. How many millilitres of blood were taken at a time? From which vein? Did you put a jugular catheter? Was it the tail vein? Preparation?

‘(10 mL by jugular venepuncture following an alcohol wipe)’ was added after Blood samples…

Line 132: was the assessor blinded? Needs to be specified?

‘who had been blinded to the treatments’ was added after ‘same person’.

Line 135: a reference for the ethogram?

This was already present and was Petherick et al (2013) – ref 24 in original submission and 19 in revised version.

Line 152: “was” should be “were”.

This was referring to behavioural variables so we think that ‘were’ is correct.

Table 2: but for the title, there is no explanation regarding the table itself. What is the table about? Remember that Tables and Figures should be able to stand alone.

The following was used as the legend for the Table: ‘Effects of treatment group (placebo, n=10; TD ketoprofen, n=10; IM ketoprofen, n=10; sham, n=6) on calves’ BW changes. Values in rows are means, which were analysed using ANOVA, with the outcomes for treatment, time and treatment * time indicated.’

Figure 1: Do you need two figures here? I believe 1b is clear enough to present your data.

The emphasis was on the first 4 hours when most (significant changes occurred) and this was consistent with the literature, but we thought that the reader would also appreciate how this changed over time – the time (x axis) squashes the first 4 hours considerably in graph b, so we considered that both graphs would be more informative.

Line 176: 96 hours not weeks

Changed

Line 179: Table 2 should be table 3

Changed

Table 3: same comment as for Table 2

The following was used as the legend for Table 3: ‘Effects of treatment group (placebo, n=10; TD ketoprofen, n=10; IM ketoprofen, n=10; sham, n=6) on calves’ plasma cortisol concentration changes. Values in rows are means, which were analysed using ANOVA, with the outcomes for treatment, time and treatment * time indicated.’

Line 180: “variables”. Shouldn't it be “behaviours” for consistency?

We thought that variables appeared correct but have changed to ‘behaviours’ to avoid confusion. 

Line 182 – 185: I understand what you mean, but the sentence is difficult to read. Is there any way it could be simplified?

This entire paragraph was re-written, as follows: ‘For the behavioural analysis, seven behaviour variables (ear flick, tail wag, ruminating, head shake, lying down, grooming and neck extending) had substantial loading and were statistically significant (Figure 2). Animal with a pain score that was higher than the groups’ average had a higher Ear flick count (0.78 Standard Deviation [SD] higher than the average for the group), higher Tail wag (0.66 SD higher), lower Rumination (- 0.36 SD) and lower in Grooming (- 0.25 SD). There was also a negative correlation between Tail wagging and Lying down, and between Ear flick and Head rub, confirming that the model framework was sound and reliable. Overall, the mean counts of positively associated pain behavioural variables in the TD group were reduced by a factor of 66% (Incident Risk Ratio [IRR] = 0.66 95% CI 0.47 to 0.92; P = 0.02), compared to placebo and sham group calves. The risk of higher mean counts of positively associated pain variables was observed at 2 hr and 4 hr after dehorning.’

Lines 187, 189 and 191: Placebo or placebo? Please be consistent throughout the paper

placebo was used (unless as a Title in a table or starting a sentence)

As a general comment, I have the impression that the interpretation of the statistical results has been written by a statistician, not a clinician. I found it very difficult to read and to understand. It would be useful to develop the interpretation (using examples) to increase comprehension.

The statistical results section was written by a statistician, since the analysis and interpretation, particularly on the behavioural data, was quite complex. The statistician re-wrote the section above to make it simpler and used examples.

Figure2: same comments as for tables 2 and 3.

The legend was re-written, as follows: ‘Line plot showing Structural Equation Modelling and Confirmatory Factor Analysis conceptual map depicting the association between calves’ behavioural variables (fitted as measured predictor variables) and pain response (fitted as a latent variable).’

Discussion

Line 198: but for cortisol, BW and behaviours, what else did you measure? That sentence implies you measured something else? I could not find what else? Please rewrite.

‘In addition’ was added to the next sentence, since comparable plasma drug concentrations is also an important comparator of efficacy (this is the basis of bioequivalence testing).

Line 201: a reference for the pilot study is missing

added

Line 203-204: The authors refer again to results from the pilot study. It is not clear at all. We need more explanation and reference.

The basic pharmacokinetic results were included at the start of the Materials and methods and the pilot study is also referenced, which is a link to the MLA website for the full report. What is intended but is unfortunately not complete is a further paper detailing a full bioavailability study (comparing IV, IM and TD) that was performed. We are happy to provide the draft to date, but it is not yet ready for submission.

Line 212-225: Paragraph on cortisol. It is a very poorly written paragraph. The readers need a better description of the curve for cortisol in other studies, followed by a comparison with the current study and a conclusion.

While we are trying to improve this manuscript and make it more readable, reviewers 2 and 3 both commented that it was a well-written paper, so a comment of very poorly written becomes a challenge to address without possibly making the intent unclear to the other reviewers.

Line 227 to 235: Very poorly written paragraph. I do not understand what the authors are demonstrating. First of all, to prove that study from McMeekan et al. (1998) is comparable, the authors should explain that it was done in calves of the same age with a similar technique (scoop). Second, if the authors had paid attention to that study, they would have realised that McMeekan at eal did not look at any behaviours, but only at the plasma concentration of cortisol. Therefore, Line 227-228 is misleading.

Apologies, the incorrect reference was used and Duffield et al (2010) is the appropriate reference (#11).  The sentence was also re-written from:

‘An early study reported that ketoprofen appeared to offer only a small reduction in adverse behaviours [11].’

To:

‘An early study reported that ketoprofen appeared to offer some reduction in adverse behaviours in calves with cornual nerve blocks dehorned by heat cauterization, compared to placebo [11].’

Line231: why is the power not part of the Mat & methods and Results sections?

We appreciate the reviewer’s comment. However, this is a post hoc analysis of the study power and therefore better placed in the discussion section.

Line 229 to 230: if I am correct, the ethogram consists of several observations and is therefore depending on the observers’ interpretations. Subjectivity is, consequently, a possible issue. The authors say that their adapted ethogram provides objective values. I don’t think this is correct as no validity tests were performed to prove this (i.e. interrater; interrater). It would have been better if more than one person had assessed the behaviours.

We agree (and agreed to a similar point above) that we could have done the behavioural interpretation better, but also note that the focus of the study was to determine if a topically-applied formulation could provide analgesia and we therefore relied on several parameters to assess efficacy on the understanding (at least from us) that there was reasonable subjectivity of some commonly-used parameters to specifically assess pain. 

Line 233-235: This sentence does not belong here.

As noted earlier, post hoc analysis was performed to determine the power of the study and we believe that the discussion is an appropriate place to state that ketoprofen analgesia did indeed provide (some) analgesia

Line 237 – 243: very poorly written paragraph.

This is difficult to address since the two other reviewers stated that the manuscript was well written.

Line 240-241: this statement is only true for Sham, TD and IM, not for the placebo group.

‘or lack thereof’ was added after ‘BW’

Line 245-247: Very poorly written. It needs rewriting. Moreover, the authors are again misleading the readers with, in this case, the study from Sutherland and Mellord (2002, ref #7). I agree that the authors of that study looked at central vs peripheric effects. But they concluded that most of the effect of ketoprofen were through the anti-inflammatory effect in the periphery and that the central effect, if any, was of very little importance. It is the opposite of what the authors in the current study seem to imply.

The intent was not to confuse or mislead, but suggest that in addition to the anti-inflammatory action of ketoprofen, there may be additional analgesic efficacy.  Indeed, the discussion in Surtherland et al (2002) states: ‘The observations in the present study therefore suggest that the delayed cortisol response may not be elicited mainly by ketoprofen-sensitive features of inflammation-related pain, as speculated by McMeekan et al (1998b), and may instead be caused by other factors as well. What they are is not clear, but they might include a transient period of central sensitisation (Woolf 1994) which begins with the onset of noxious sensory input from the amputation wounds once the effects of the local anaesthetic wear off.’

To hopefully avoid confusion, we re-wrote the section from:

‘It could be expected that the use of a NSAID would ameliorate the physiological and behavioural responses to surgical pain.  The initial pain response may be unchanged but certainly cortisol responses are not as pronounced by 2 hr after dehorning in ketoprofen is used [McMeekan et al 1998]. Interestingly, a similar response was not found when phenylbutazone was administered, suggesting a potential central effect of ketoprofen [Sutherland et al 2002].’

To:

‘It could be expected that the use of a NSAID would ameliorate the physiological and behavioural responses to surgical pain.  The initial pain response may be unchanged but certainly cortisol responses are not as pronounced by 2 hr after dehorning if ketoprofen is used [Duffield et al 2010]. Interestingly, a similar response was not found when phenylbutazone was administered, suggesting a potential central effect of ketoprofen, in addition to the primary anti-inflammatory effect [Sutherland et al 2002].’

Line 255-259: This should be part of the results section.

We disagree – the reviewer seems to be referring to allo-grooming not contributing to systemic drug concentrations (lines 255-257), while the reference to limited statistical power (lines 257-259) is to suggest that more animal numbers may have made some of the behavioural changes observed significant – we think this appears reasonable to include in the discussion?

Line 245-259: very poorly written. It needs to be developed better and rewritten.

This is difficult to address since the two other reviewers stated that the manuscript was well written.

Conclusion:

While info on line 261 could be extrapolated from the results of this study, line 262-263 was not part of the study. Nowhere in the study, we looked at the facility of providing the TD to cattle, even fractious ones. It is pure speculation.

‘A transdermal formulation containing ketoprofen was able to attain rapid therapeutic plasma drug concentrations’.

Was changed to:

‘A transdermal formulation containing ketoprofen appeared to provide analgesia in calves undergoing dehorning as assessed by several parameters’

In summary, the discussion section lacks of information on the different types of dehorning methods and how they compare with the one used in the study. There is no discussions about the different ways of assessing pain in calves, objective vs subjective and how it compares with this study.

There is no real discussion of the weakness of the papers: the validity of the assessment method, the use of only one observer etc.

We have responded above that we were certainly not trying to repeat much of the excellent work done to date on assessing the pain response to various types of dehorning and none of us are behaviour specialists (We tried to get Carol Petherick involved, but she retired!). The title of the manuscript begins: ‘A Novel Transdermal Ketoprofen Formulation…’ and this is the first time a topically-applied formulation (allowing for the sparse information available about the flunixin formulation registered overseas for pyrexia) has been shown to have efficacy.  This efficacy could have relied on a comparison with IM alone (since comparable plasma drug concentrations are accepted to support comparable efficacy), but we wished to include some of the traditional parameters to compare with previous studies. 

In summary:

I think you have a good study that deserves to be published. Still, the authors must rework on the introduction, the presentation of the results and most notably on the structure and development of the discussion.

We hope this is now satisfactory

My main concern about your results (and the design of the study) is that you only used one assessor. You should have filmed the calves and have different people assessing the behaviours. It would have given you better quality data. Moreover, you could have used the results to validate your assessment method partially.

We agree, but this was quite an intensive study and we did not have access to sufficient cameras, facilities and staff to undertake what the reviewer is suggesting. The authors consider that a major outcome was successful transdermal delivery of a NSAID that has scientific and producer acceptance as an analgesic AND that it was non-inferior to IM ketoprofen. However, we are very keen to see more widespread use of this formulation and possible commercialisation if we can attract a pharma partner, so further studies and more involved efficacy studies are planned.

Reviewer 2 Report

Reviewer comments on Manuscript number: animals-1022559

The present manuscript shows the analgesic effect of a transdermal ketoprofen formulation in calves undergoing amputation dehorning.

The manuscript is well written, and the information showed is interesting and relevant, as its main aim is to provide an analgesic alternative to a common husbandry practice.

Broad comments

The study is well designed, and the data showed is valuable, however, some parts of the manuscript need major clarification before publication.

Weakness of the study: A higher expected CMAX in transdermal formulation than an intramuscular administration. Lack of providing local anesthesia for trans-dehorning acute pain management or even light sedation for chemical restraint. Extremely variation on serum cortisol levels at 0 hours. Number of animals per group.

Specific Comments

Abstract

Line 3. I recommend to change “calves” instead cattle, as it better describes the subject of the study.

Line 32. Please specify those pain variables.

Lines 36-37. Pain relief cannot be administered, only drugs. Please rephrase.

Lines 46-47. Keywords. Please add calf.

Material and methods

Line 118-119. It is not clear the way you dosed the transdermal ketoprofen. How much did you apply considering the weight of the calf? Please explain and add this missing important information.

Line 120. It would be better to specify the anatomical area and its relevant muscles rather “rump”.

Line 122: Please briefly describe the method of dehorning.

Line 132. Please indicate if that single person was exactly the same who did all the monitoring of calf behavior during the entire project, and please specify the time between each dehorning was performed.

L 139. What was the plan if the calf licked the site of application? Did they do it?

Results

L 176. You meant 96 hours rather than weeks right? Please precise it.

L 179. Concentration.

There is a wide range of plasma cortisol values at time 0 that may biased your results. Placebo had 10.2 and raise to 22.5 after an hour and IM ketoprofen had 2.6 and raised almost two-fold at the same time. This results need further discussion.

Discussion

L 201-204. The aim of this manuscript did not include results of ketoprofen serum concentrations, then this paragraph should be removed from this section.

L 206-208. To this reviewer, all the treated and non-treated animals (probably excluding sham animals) should have received a corneal local block, considering animal welfare; however I understand that the main idea is to avoid the participation of a DVM and to facilitate the administration of pain medication before the dehorning.

L 210. What were the expected adverse side-effects? Please specify.

It is important to discuss the probability that a higher expected Cmax of ketoprofen after topical administration would have a clearly dose-dependent analgesic effect and probably side-effects when comparing to IM.

L 241-242. What is the cost of each treatment?

L 248. Ketoprofen has a dual anti-inflammatory effect by acting on cyclooxygenases and Lipoxygenases. A potential central effect could not be explained by your results.

Conclusions

They must be re-written considering your results and the aim of the study. You did not measure ketoprofen plasma drug concentrations.

Thank you.

Author Response

A novel transdermal ketoprofen formulation provides effective analgesia to cattle undergoing amputation dehorning – response to reviewer’s comments

Reviewer #2

The present manuscript shows the analgesic effect of a transdermal ketoprofen formulation in calves undergoing amputation dehorning.

The manuscript is well written, and the information showed is interesting and relevant, as its main aim is to provide an analgesic alternative to a common husbandry practice.

Broad comments

The study is well designed, and the data showed is valuable, however, some parts of the manuscript need major clarification before publication.

Weakness of the study:

A higher expected CMAX in transdermal formulation than an intramuscular administration.

A generalisation from transdermal drug delivery is that a dose rate of 10 X or more is required to attain enteral/parenteral systemic concentrations. With the TD ketoprofen, a dose rate ~ 3X was used, although subsequent pharmacokinetic studies revealed that 1-1.5X would be sufficient.  Importantly, the plasma drug concentrations persisted for longer, compared to IM, which would certainly contribute to prolonged analgesia.  The major outcome was that a transdermal formulation could attain therapeutic plasma drug concentrations and this could easily be brought closer to the IM plasma concentrations (if this is indeed what is preferred, although, as already noted, the extended ‘tail’ would contribute to prolonged efficacy following TD).  It was not an intent to specifically compare routes of delivery of ketoprofen. Compliance in the use of analgesia would certainly increase if administration was simple and could be performed by non-veterinary staff.

Lack of providing local anesthesia for trans-dehorning acute pain management or even light sedation for chemical restraint.

This was a discussion point before we received approval from the UQ Animal Ethics Committee.  The approval was provided on the basis that: (1) not using local anesthesia was industry standard (i.e. not always or usually provided, particularly on extensive properties); (this has been added at line 100-101) (2) The ketoprofen was active (i.e. providing analgesia) before the dehorning occurred, with NSAIDs, including ketoprofen, shown to provide effective analgesia in the literature.  Since it was applied in the race and time elapsed while preparing to dehorn, this would provide pre-emptive analgesia, whereas ketoprofen, if administered in practice, would normally be provided just after the procedure. 

Extremely variation on serum cortisol levels at 0 hours.

Cortisol is known for its variability, particularly in species such as cattle, and it was not intended to be the major indicator of efficacy but had been included in similar previous studies.  A multifactorial approach was used in the current study, including cortisol, plus body weight changes and behaviour, to provide a balanced measure of efficacy.  However, limited emphasis was placed on cortisol due to its known variability and lack of objectivity (i.e. there were some increases in plasma cortisol in the Sham (non-dehorned group)).  

Number of animals per group

More animals are always useful in a study, but we ran with the numbers indicated in response to the UQ Animal Ethics committee strongly advocating for minimal numbers as part of a ‘reduce, replace, refine’ approach. It was also difficult to effectively monitor behavioural changes if more animals had been included with the staff available. Irrespective, significant differences were found were found in several key parameters, which we considered to justify the animal numbers included.

Specific Comments

Abstract

Line 3. I recommend to change “calves” instead cattle, as it better describes the subject of the study.

Done

Line 32. Please specify those pain variables.

(ear flick, tail wag, ruminating, head shake, lying down, grooming and neck extending) was added

Lines 36-37. Pain relief cannot be administered, only drugs. Please rephrase.

‘Pain relief’ was changed to ‘analgesic drugs’

Lines 46-47. Keywords. Please add calf.

done

Material and methods

Line 118-119. It is not clear the way you dosed the transdermal ketoprofen. How much did you apply considering the weight of the calf? Please explain and add this missing important information.

This was not intended as a pharmacokinetic study or even a bioavailability study, but to reflect what may occur in practice, particularly with large numbers of relatively unhandled animals. So while the animals were weighed, a standard dose rate (~ 10 mL) was applied to each calf when several had been confined in a race. If this formulation becomes commercialised, it would likely be applied by a lay handler using a pour-on style of applicator. The sentence ‘A standard volume of 10 mL was applied to each calf’ was added.

Line 120. It would be better to specify the anatomical area and its relevant muscles rather “rump”.

Rump was replaced by gluteus medius muscle

Line 122: Please briefly describe the method of dehorning.

A description has been added to the text (lines 121-130).

Line 132. Please indicate if that single person was exactly the same who did all the monitoring of calf behavior during the entire project, and please specify the time between each dehorning was performed.

The following was already in the text: ‘Calf behaviour was monitored before and after dehorning by the same person. The day before dehorning, the behaviour of all calves were monitored for two hours to record the baseline variation, which was taken as 0 hour observation. After dehorning, behaviour was monitored at 2-4 hr, 4-8 hr, 8-12 hr, at 24 hr and at 48 hr. A behaviour ethogram adapted from Petherick et al (2013) was used for all observations. Twelve behaviours were monitored: head shaking, ear flicking, tail wagging, head rubbing, lying, ruminating, neck extending, grooming, walking, vocalising, feeding and drinking. Each animal was observed for 3 min at each observation period and the frequency of each behaviour recorded.’ (although we have now changed ‘..a single person’ to ‘…the same person).

The time between dehorning for animals within the race was 2-3 min and a little longer (~ 5 min) when additional animals were encouraged into the race as the last animal previously in the race was being dehorned.

L 139. What was the plan if the calf licked the site of application? Did they do it?

The plan was to discourage them (by walking towards them) but no animal exhibited any intention to lick, at least for the first12 hr of the study.

Results

L 176. You meant 96 hours rather than weeks right? Please precise it.

Correct and this has been amended.

L 179. Concentration.

Corrected.

There is a wide range of plasma cortisol values at time 0 that may biased your results. Placebo had 10.2 and raise to 22.5 after an hour and IM ketoprofen had 2.6 and raised almost two-fold at the same time. This results need further discussion.

As we noted above, cortisol is known for its variability, particularly in species such as cattle, and it was not intended to be the major indicator of efficacy.  However, we have added into the Discussion section (paragraph 3) the following: ‘One of these concerns is the high variability in resting cortisol levels and the response to stress. For example, mild stress (confining in a pen) induced significant differences in the cortisol responses within a group of nine Angus/Hereford cows [21]. It is therefore prudent to use other indicators of stress and, particularly for the current study, pain when monitoring bovine responses.’

Ref 21 is: Bristow DJ, Holmes DS (2007). Cortisol levels and anxiety-related behaviors in cattle. Physiol Behav 90(4), 626-628.

Discussion

L 201-204. The aim of this manuscript did not include results of ketoprofen serum concentrations, then this paragraph should be removed from this section.

This statement was focussed on the results of the pilot study, which we did mention in the Materials and Methods, where we also referred to the study report [15].  To clarify this, we added ‘…from the pilot study’ to line 210. 

L 206-208. To this reviewer, all the treated and non-treated animals (probably excluding sham animals) should have received a corneal local block, considering animal welfare; however I understand that the main idea is to avoid the participation of a DVM and to facilitate the administration of pain medication before the dehorning.

We agree that a cornual nerve block would be appropriate for dehorning cattle.  However, this would have obscured the analgesic effects of ketoprofen (IM or TD), which was the main point of the study.  As noted earlier, we did receive permission to not use a cornual nerve block from the UQ Animal Ethics Committee, which included the understanding that dehorning may be performed without a cornual nerve block (or indeed any analgesia) in extensive properties well outside urban regions.  This lack of analgesia (due to proximity to suitably trained personnel, such as a DVM, and cost) was the reason we developed the TD formulation originally.  It should also be noted that, with the rapid transdermal penetration (or following IM administration), ketoprofen was present systemically when the actual dehorning was performed.

L 210. What were the expected adverse side-effects? Please specify.

It is important to discuss the probability that a higher expected Cmax of ketoprofen after topical administration would have a clearly dose-dependent analgesic effect and probably side-effects when comparing to IM.

This comment goes against the comment in the Conclusions section to re-write the section since we did not measure ketoprofen plasma drug concentrations.  It is also pharmacokinetically inadvisable to state that increasing the dose will necessarily increase the analgesic effect.  Dose-dependent analgesia will apply to all NSAIDs at lower dose rates, whereas at higher dose rates, this is less likely to apply as maximum efficacy is approached (which is likely what the IM dose rate is intended to approach). However, as the reviewer notes, increasing dose rate will certainly increase the likelihood of adverse effects, although this is unlikely for NSAIDs from a single dose (and we suggested lowering the TD dose rate for future studies).  We were also unsure if the reviewer was specifically referring to adverse drug effects (which can range from common effects following NSAIDs, such as gastrointestinal bleeding and renal dysfunction) to a systemic reaction to the drug (or vehicle) itself or to the dehorning procedure. To address these concerns, after: ‘no adverse effects were observed’ we have added: ‘either from the procedure or the drug administration. There was potentially a higher risk of adverse effects commonly associated with NSAIDs (i.e. renal, gastrointestinal and haemostatic) following TD administration due the higher maximum plasma concentration, compared to IM, although this was considered minimal since only a single dose was used.’

L 241-242. What is the cost of each treatment?

It cost ~ 20 – 30 cents (AUS) a dose.  It was essential that the final formulation was comparatively cheap, otherwise it would not be viable to a pharmaceutical company to manufacture or for a producer to use.

L 248. Ketoprofen has a dual anti-inflammatory effect by acting on cyclooxygenases and Lipoxygenases. A potential central effect could not be explained by your results.

The dual anti-inflammatory effect of ketoprofen appears to be uncertain, with Plumb (Veterinary Drug Handbook) stating that: ‘ketoprofen has purportedly an anti-lipoxygenase activity, but this has not been demonstrated in vitro using samples from several species’. Furthermore, a study reported that ketoprofen did not inhibit 5-lipoxygenase or leukotriene B4 (Salman, et al. (1984). Biochem Pharmacol. 33: 28–2930).  Irrespective, the comment in the text reflected a direct statement from Sutherland et al (2002) that ketoprofen appears to provide an early (within 2 hr) analgesic effect that is not observed with phenylbutazone. These authors attributed this superior efficacy to a possible central effect of ketoprofen, which is where the statement in the text came from (and was cited).  It was also noted by Stafford and Mellor (2005) in their review of analgesia in calves that: ‘The effect of ketoprofen in preventing the establishment of the cortisol response associated with inflammation-related pain may be due to either its peripheral or central effects, or both’

Conclusions

They must be re-written considering your results and the aim of the study. You did not measure ketoprofen plasma drug concentrations.

To avoid confusion, ‘(pilot study data [15])’ was added to the first sentence in the conclusion, while ‘comparative IM plasma concentrations’ was changed to ‘comparative responses to IM ketoprofen administration’.

Reviewer 3 Report

Thank you for submitting this well written and succinct manuscript.  I have just a few comments:

  • line 60 - include reference to the optimal timing of administration of NSAIDs prior to a surgical procedure
  • Methods - provide information on the fate of the animals after the study
  • Figure 2 - correct typographical errors in ruminating, grooming and neck extended

Author Response

A novel transdermal ketoprofen formulation provides effective analgesia to cattle undergoing amputation dehorning – response to reviewer’s comments

 Reviewer #3

Thank you for submitting this well written and succinct manuscript.  I have just a few comments:

  • line 60 - include reference to the optimal timing of administration of NSAIDs prior to a surgical procedure

This is an interesting question, since many NSAIDs and administered after injury or inflammation has occurred, so most dosing is based on achieving a rapid systemic concentration. Pre-emptive analgesia is however widely recognised as providing superior analgesia, since the presence of the analgesic agent, particularly an NSAID, is already present when the inflammatory mediators are induced. Unfortunately, pre-emptive NSAIDs are still controversial with anaesthetists since they may have systemic effects during the procedure, including affecting renal function and haemostasis. Irrespective, Stafford and Mellor are widely recognised for their expertise on analgesia in ruminants and their 2005 review suggest ketoprofen should be administered IV 15-20 min before a surgical procedure. The TD formulation achieves therapeutic (bit not peak or Cmax) concentration ~ 20 min after application, so application ~ 30 min prior to dehorning would appear reasonable. The following sentence was added to the introduction: ‘Stafford and Mellor [5] suggested that administration of ketoprofen intravenously at 15-20 min prior to the surgical procedure may be appropriate.’

  • Methods - provide information on the fate of the animals after the study

The calves were returned to the UQ Dairy Research and Training herd once the study was completed.

  • Figure 2 - correct typographical errors in ruminating, grooming and neck extended

Round 2

Reviewer 1 Report

Dear Authors,

After reading your response to my comments, I decided to make 2 comments:

  • You might indeed have found my comments hard, but I intended to make your paper stronger. I agree that I sometimes tend to review papers as I would do for my team.
  • As a result, I decided to read your latest version without looking at your specific response to see how it reads now, trying not to be influenced by our comments.

Simple summary

Very good

Abstract

Line 29: Not sure if it is because of the PDF format, but the space between “N=” and “6” is bigger than previous ones (N=10)

Introduction

Line 49: Ref # 5 (line 320-321. The title is incorrect: “Pain mechanisms and their implication for the management of pain in farm and companion animals.”

Material & Methods

Line 122: “The operator was blinded from the treatment”. I am a bit confused here. Was the Operator blinded to all treatments of just TD and Placebo? If all treatment, I would assume therefore that all the calves received the Placebo treatment, but for TD, that all the calves received an IM injection (saline) but for the IM group etc. Please could you specify? On line 145, the authors mention that the Operator was also the pain assessor. Therefore, the readers need to know if that person was blinded for all treatment or just the TD keto and TD Placebo. If the latter, that needs to be addressed in the discussion.

Line 173: Line should not start with an abbreviation

Results

Line 175: “bodyweight” or BW?

Line 187: Table 2

The table should stand alone, as does Table one.

For example:

“Effects of treatment group on calves’ BW changes at Days 0, 14 and 35 after dehorning in cattle treated with placebo (n=10), TD ketoprofen (n=10; 10 mg/kg), IM ketoprofen (n=10; 3 mg/kg), compared to a sham group that were not dehorned (n=6).”

Line 196: Legend for Figure 1 is good. I still don’t think that you need figure 1a. Figure 1 b is clear enough. I understand why you do it, but I don’t think it makes a difference.

Could you check the size used for figure 1 a and figure 1 b? The “a” and the “b”, as well as the word “cortisol”, are in different sizes (consistency).

Line 199: Same comments as fo the legend of Table 2. It needs to stand alone. The legend does not mention dehorning.

Discussion:

Line 281: “…outcome from the current was…” it seems a word is missing. “study”, maybe?

Thank you

Author Response

After reading your response to my comments, I decided to make 2 comments:

   You might indeed have found my comments hard, but I intended to make your paper stronger. I agree that I sometimes tend to review papers as I would do for my team.

    As a result, I decided to read your latest version without looking at your specific response to see how it reads now, trying not to be influenced by our comments.

Simple summary

Very good

 Many thanks and really appreciated

Abstract

Line 29: Not sure if it is because of the PDF format, but the space between “N=” and “6” is bigger than previous ones (N=10)

corrected

Introduction

Line 49: Ref # 5 (line 320-321. The title is incorrect: “Pain mechanisms and their implication for the management of pain in farm and companion animals.”

corrected

Material & Methods

Line 122: “The operator was blinded from the treatment”. I am a bit confused here. Was the Operator blinded to all treatments of just TD and Placebo? If all treatment, I would assume therefore that all the calves received the Placebo treatment, but for TD, that all the calves received an IM injection (saline) but for the IM group etc. Please could you specify? On line 145, the authors mention that the Operator was also the pain assessor. Therefore, the readers need to know if that person was blinded for all treatment or just the TD keto and TD Placebo. If the latter, that needs to be addressed in the discussion.

This has been corrected to indicate that the person monitoring behaviour was blinded rto he treatments.

Line 173: Line should not start with an abbreviation

corrected

Results

Line 175: “bodyweight” or BW?

corrected - BW was used unless starting a sentence

Line 187: Table 2

The table should stand alone, as does Table one.

corrected

For example:

“Effects of treatment group on calves’ BW changes at Days 0, 14 and 35 after dehorning in cattle treated with placebo (n=10), TD ketoprofen (n=10; 10 mg/kg), IM ketoprofen (n=10; 3 mg/kg), compared to a sham group that were not dehorned (n=6).”

Corrected as suggested

Line 196: Legend for Figure 1 is good. I still don’t think that you need figure 1a. Figure 1 b is clear enough. I understand why you do it, but I don’t think it makes a difference.

Thank you – this figure was included following the advice of our statistician

Could you check the size used for figure 1 a and figure 1 b? The “a” and the “b”, as well as the word “cortisol”, are in different sizes (consistency).

This figure was re-drawn and is now consistent

Line 199: Same comments as fo the legend of Table 2. It needs to stand alone. The legend does not mention dehorning.

corrected as suggested

Discussion:

Line 281: “…outcome from the current was…” it seems a word is missing. “study”, maybe?

‘study’ was added

Thank you

Reviewer 2 Report

Dear authors:

Thanks for improving your manuscript; however this reviewer still thinks that the conclusion does not reflect the aim of your study.

Your aim:

The objective of the current study was to determine whether effective analgesia was induced following topical application of a transdermal formulation of ketoprofen prior to amputation dehorning of calves

Your conclusion:

"A transdermal formulation containing ketoprofen was able to attain rapid therapeutic plasma drug concentrations(pilot study data [15]).This formulation can be readily applied by lay (non-veterinary) operators even to fractious animals, with the rapid onset of analgesia (efficacy judged by comparative responses to IM ketoprofen administration) ideal for surgical husbandry procedures, particularly in extensive industries"

The pilot study was not part of your current study, then you can't conclude about therapeutic plasma drug concentrations. You should conclude about comparative analgesia  (TD vs IM), and  its effects on weight gain.

Comments and /or suggestions should not be part of the conclusions.

Thank you

Author Response

We have changed the conclusion from:

A transdermal formulation containing ketoprofen was able to attain rapid therapeutic plasma drug concentrations(pilot study data [15]).This formulation can be readily applied by lay (non-veterinary) operators even to fractious animals, with the rapid onset of analgesia (efficacy judged by comparative responses to IM ketoprofen administration) ideal for surgical husbandry procedures, particularly in extensive industries

to:

Administration of a transdermal formulation containing ketoprofen to calves prior to amputation dehorning resulted in a rapid onset of analgesia, very similar in duration and efficacy to that induced by IM administration of ketoprofen. Because of the recognised ease of administration of transdermal formulations to cattle these findings are likely to facilitate greater routine use of analgesia in cattle undergoing painful husbandry procedures such as dehorning.